# A Comparative Study of Brain Injury Biomarker S100β During General and Spinal Anesthesia for Caesarean Delivery: A Prospective Study

**DOI:** 10.3390/medicina61081382

**Published:** 2025-07-30

**Authors:** Mungun Banzar, Nasantogtokh Erdenebileg, Tulgaa Surjavkhlan, Enkhtsetseg Jamsranjav, Munkhtsetseg Janlav, Ganbold Lundeg

**Affiliations:** 1Department of Critical Care Medicine and Anesthesiology, School of Medicine, Mongolian National University of Medical Sciences, Ulaanbaatar 14210, Mongolia; mungunbanzar@gmail.com (M.B.); ganbold@mnums.edu.mn (G.L.); 2Departmentof Anesthesiology, National Center for Maternal and Child Health, Ulaanbaatar 16060, Mongolia; 3Research Department, National Center for Maternal and Child Health, Ulaanbaatar 16060, Mongolia; nasantogtox.e@gmail.com; 4Department of Biochemistry, School of Biomedicine, Mongolian National University of Medical Sciences, Ulaanbaatar 14210, Mongolia; tulgaa@mnums.edu.mn; 5Department of Obstetrics and Gynecology, School of Medicine, Mongolian National University of Medical Sciences, Ulaanbaatar 14210, Mongolia; enkhtsetsegjamsaranjav@gmail.com

**Keywords:** S100β, Cesarian section, Anesthesia

## Abstract

*Background and Objectives*: Anesthetic agents may influence brain function, and emerging evidence suggests possible neurotoxicity under certain conditions. S100β is a well-established biomarker of brain injury and blood–brain barrier disruption, and its prolonged elevation beyond 6–12 h, despite a short half-life, may indicate ongoing neuronal injury. Its use in cesarean section (C-section) remains limited, despite the potential neurological implications of both surgical stress and anesthetic technique. This study evaluates potential brain injury during caesarean section by comparing maternal and neonatal S100β levels under general and spinal anesthesia. *Materials and Methods*: This observational prospective study compared changes in the S100β brain damage biomarker in maternal (pre- and post-surgery) and umbilical artery blood during elective c-sections under general or spinal anesthesia. The 60 parturient women who underwent a C-section from 1 July 2021 to 30 December 2023 were evenly distributed into 2 groups: General anesthesia (GA) (n = 30) and Spinal anesthesia (SA) group (n = 30). It included healthy term pregnant women aged 18–40, ASA I–II and excluded those with major comorbidities or emergency conditions. *Results*: S100β concentrations slightly increased once the C-section was over in both the SA and GA groups, but without notable differences. In the SA and GA groups, preoperative S100β concentration in maternal blood was 195.1 ± 36.2 ng/L, 193.0 ± 54.3 ng/L, then increased to 200.9 ± 42.9 ng/L, 197.0 ± 42.7 at the end of operation. There was no statistically significant difference in S100β concentrations between the spinal and general anesthesia groups (*p* = 0.86). *Conclusions*: S100β concentrations slightly increased after C-section in both groups. The form of anesthesia seems to be irrelevant for the S100β level. However, further research is needed to confirm these findings and fully evaluate any potential long-term effects.

## 1. Introduction

In the obstetrics department, caesarean section is the most common surgery. The World Health Organization (WHO) research shows that global C-section delivery has risen to over 21% of all childbirths [1]. According to a study by the Center for Health Development, the C-section rate in Mongolia has been increasing over the years from 22.4% in 2012 to 25.4% in 2016 [2]. Anesthesiologists must decide whether general or regional anesthesia is optimal for cesarean section delivery based on the specific condition and clinical circumstances of each patient.

A classical method of anesthesia for C-section delivery is spinal anesthesia (SA), and through relevant and rigorous guidelines can be recommended to patients. Spinal anesthesia is rapid, providing quick onset of bilateral, dense, and reliable anesthesia using minimal drug dosages with low risk of both material toxicity and fetal drug transfer. One drawback of spinal anesthesia is its fixed duration after a single injection. A major adverse fetal effect of SA is maternal sympathetic blockade, resulting in uteroplacental hypoperfusion, causing hypotension and a decrease in the intervillous blood flow, potentially leading to acidemia [3,4,5]. GA provides a rapid and reliable onset, with steady control over the airway and ventilation, potentially reducing the risk of hypotension. GA may be more suitable in certain conditions (e.g., profound fetal bradycardia, ruptured uterus, severe hemorrhage, severe placental abruption, umbilical cord prolapse, and preterm footling breech) [6]. Traditionally, there was a belief that exposure to general anesthesia during a cesarean section could lead to birth asphyxia [7]. However, the current understanding suggests that brief exposure to GA is generally safe for the fetus. Short-acting anesthetics like propofol are preferred to minimize exposure time and reduce potential risks [8]. Propofol offers a rapid, reliable loss of consciousness [9]. It is an intravenous sedative-hypnotic agent, containing amnestic properties for the induction of GA. It also lowers cerebral blood flow, intracranial pressure, and cerebral metabolic rate whilst preserving static autoregulation [10] and vascular responsiveness to carbon dioxide [11]. Propofol’s neuroprotective effects are thought to stem from its antioxidant capabilities, its enhancement of γ-aminobutyric acid (GABAA)-mediated inhibition of synaptic transmission, and its suppression of glutamate release in cerebral ventricles [12]. On the other hand, GA has been demonstrated to be neurotoxic for certain animals’ developing brains in a dose and time manner and may be associated with both long-term learning and development disorders [13]. However, these effects of GA cannot be confirmed for the human species. The mechanisms underlying general anesthesia-mediated effects, including neuroprotection and neurotoxicity, remain unclear despite various proposed hypotheses. The development of a reliable biomarker to detect acute anesthesia neurotoxicity in the brain could significantly enhance research progress. However, uncertainties persist regarding its effectiveness in reflecting anesthesia-related brain injury in humans. The choice of anesthetic technique is often guided by clinical circumstances and patient factors, yet emerging evidence suggests it may also have significant implications for neuroinflammatory responses and glial activation. Understanding the neuroglial impact of different anesthetic modalities is especially important given their potential to influence postoperative cognitive outcomes, neurodevelopment, and neurodegeneration. By elucidating how general and regional anesthesia affect neuroimmune signaling, this study aims to inform safer anesthetic practices and identify modifiable risk factors relevant to perioperative brain health [14]. Several neuroinflammatory markers—such as S100B, neuron-specific enolase (NSE), glial fibrillary acidic protein (GFAP), tumor necrosis factor-alpha (TNF-α), and interleukin-6 (IL-6)—have been investigated or are hypothesized to vary in response to general anesthesia (GA) and spinal anesthesia (SA), particularly in pregnant women undergoing cesarean section. Among these markers, S100B has been most frequently studied in relation to cesarean delivery and comparisons between anesthesia modalities. Therefore, S100β was chosen as the primary biomarker of interest in the present study [15].

The calcium-binding protein S100β serves as a widely employed biomarker for identifying brain damage resulting from various stressors, such as ischemia and trauma [16,17]. Notably sensitive, S100β, primarily found in astrocytes due to its affinity for calcium, plays a pivotal role in regulating diverse intra- and extracellular physiological processes. It functions both as a marker and a signal, activating detection and protective mechanisms upon central nervous system (CNS) damage [18]. Studies indicate S100β’s established specificity as a marker for CNS tissue damage. Its presence in biological fluids, including cerebrospinal fluid, peripheral and cord blood, urine, saliva, and amniotic fluid, signifies active neural distress [19]. Following brain injury or surgical intervention, S100β levels typically peak within 30 min to 2 h. Given its relatively short half-life (approximately 30 to 120 min), persistent elevation beyond 6 to 12 h may suggest ongoing or secondary neuronal injury [20]. This protein has been shown to be valuable in animal models for detecting general anesthesia-mediated acute neurologic damage in the developing brain and also used to monitor the impact of prenatal drug exposure. However, its effectiveness and usefulness to reflect anesthesia-associated brain damage in human subjects are yet to be confirmed [21,22].

The umbilical artery carries deoxygenated blood from the developing fetus’s circulation and demonstrates fetal changes, whereas the umbilical vein holds oxygenated blood originating in the placenta and shows changes from the mother [23]. Due to its high sensitivity in detecting fetal brain damage caused by asphyxia or prenatal stress, S100β is considered a strong candidate for investigation in related clinical studies. Some studies suggest that S100β may also serve as a biomarker for neurodegeneration in the developing brain induced by general anesthesia. This is supported by the observation that S100β levels remain consistent across umbilical artery (UA), umbilical vein (UV), and the UA/UV ratio, indicating its reliability as a peripheral biomarker in neonates [24].

In Mongolia, SA and GA are the predominant techniques used during cesarean section. However, to date, no studies have systematically examined their respective effects on maternal and fetal outcomes in this context. Utilizing the brain damage biomarker S100β, we compared the ratio of S100β levels between maternal arterial blood and umbilical artery blood immediately post-delivery, and assayed fetal S100β levels. All patients underwent a C-section using SA or GA. Our hypothesis suggests that the brain damage biomarker S100β would exhibit no elevation in the cord arterial blood of fetuses who experienced brief exposure to general anesthetics compared to those who underwent C-section using spinal anesthesia.

## 2. Materials and Methods

### 2.1. Study Design and Participants

The study was an observational, prospective, longitudinal design. In accordance with the inclusion and exclusion criteria, the parturient women who undergo C-section in the Obstetrics and Gynecology Hospital of the National Center of the Maternal and Child Health of Mongolia from July 2021 to December 2023 enrolled in this study. The terms general anesthesia (GA) and spinal anesthesia (SA) were used to designate two distinct groups. Women giving birth by caesarean section in the hospital were granted permission to take part in the study and were registered. Figure 1 shows patient recruitment and flow. The sample size was calculated based on the difference in means under the hypothesis of a difference between the two groups. A minimum of 29 subjects was required per study group. The study adhered to the following specific inclusion and exclusion criteria for the participants.

The inclusion criteria were as follows: (1) aged between 18–40; (2) American Society of Anesthesiologists (ASA) physical status I or II; (3) term gestation at 37 weeks; (4) patients whose hemoglobin > 100 g/L; and (5) women with uncomplicated singleton pregnancies who were advised to undergo elective caesarean sections due to factors such as a previous caesarean delivery, a history of primary infertility, or other reasons. Exclusion criteria were as follows: (1) less than height 150 cm; (2) body mass index (BMI) ≥ 30 kg/m^2^; (3) parturient women who suffered from severe internal, surgical, or obstetric comorbidities; (4) preeclampsia; (5) known fetal neurologic deficit, intrauterine growth retardation; (6) patients who received analgesic and sedative medicine before C-section; (7) patients who suffering from a severe mental illness; (8) emergency C-section for delivery; (9) classification as ASA status III, IV and V; (10) patients who were unwilling to partake in the study; (11) patients who were allergic to anesthetics; and (12) patients who had contraindications to general/or spinal anesthesia.

Maternal demographic data, as well as preoperative and intraoperative hemodynamic fluctuations, operative duration, blood loss, urine output, volume of infused solutions, oxytocin dosage, and postoperative symptoms (including postoperative nausea and vomiting [PONV], headache, fever, and pain), were documented.

Postpartum Apgar scores, umbilical arterial blood gas parameters, electrolyte levels, hemoglobin derivatives, incidence of neonatal asphyxia, and neonatal intensive care unit (NICU) admission data were collected for the neonates.

### 2.2. Follow-Up and Outcome Measurements

#### 2.2.1. Maternal Outcomes

*Preoperative and intraoperative*: Preoperative assessments were carried out by the main investigators and the researchers gathered demographics data and basal health questionnaires. Radial arterial cannulation was performed for all the patients using 20G Surflo^®^ (Terumo China holding Co., Ltd., Beijing, China) under local anesthesia (2% lidocaine), and arterial blood pressure was monitored. Invasive blood pressure readings, including systolic, diastolic, and mean blood pressures, were obtained. Heart rate and SpO_2_ were measured by the fingertip photoelectric sensor (manufactured: Guangzhou Sichuang Hongyi Electronic Technology Co., Ltd., Guangzhou, China). Time from anesthesia induction to delivery, total operative time, intraoperative blood loss, urine output, and complications during surgery were recorded in both groups. All vital signs were recorded.

#### 2.2.2. Postoperative

For the two hours of the postoperative stage, patients were monitored in the recovery room. Complications after C-section within two hours were recorded in both groups, including postoperative nausea, vomiting, headache, fever, and pain. The patients were assessed using the Numeric Pain Rating Scale (NPS), which showed the severity of the postoperative level of pain from a scale of 0 to 10, with 0 = no pain and 10 being the worst pain ever. The onset time was also recorded.

#### 2.2.3. Neonatal Outcomes

Following the delivery of the baby, pediatricians assessed the neonatal condition, assigned Apgar scores at one and five minutes, and determined the need for admission to the Neonatal Intensive Care Unit (NICU). An Apgar score of 7 to 10 was considered normal; 4 to 6, mild neonatal asphyxia; and 3 and below, severe neonatal asphyxia. The Apgar score was assessed based on 5 criteria: activity, pulse rate, grimace (reflex irritability), skin color, and respiratory effort. Following the delivery of the baby, the cord was cut between the two clamps placed approximately 10 to 12 cm apart and away from the placenta and newborn. One to three ml of blood was collected from the umbilical artery between the clamps immediately following placental delivery. This protocol was designed to prevent contamination of S100β levels in the cord arterial blood by placental S100β, ensuring that the measured concentrations accurately reflect those originating from the newborn.

### 2.3. Maternal and Neonatal Brain Injury Marker and Other Laboratories Outcome Measurements

Measurement of serum s100β: The maternal serum S100β protein levels of the 30 patients from each group were analyzed pre- and postoperatively. A total of 5 mL of maternal blood was withdrawn via the arterial line, while 1–2 mL of umbilical cord blood was collected from the neonates directly after birth using a 3 mL syringe. The caesarean section, performed under either spinal or general anesthesia, typically lasted approximately 50 min. Arterial blood samples were collected immediately postoperatively for the quantification of S100β protein levels. The blood samples were immediately centrifuged at 2500 RPM for 10 min and the supernatant was gathered and stored at minus 80-degree temperature until the measurement of S100β levels. The concentration of S100β protein in the blood serum was determined by the enzyme-linked antibody reaction (ELISA) using the Human S100 calcium binding protein B (S100B) ELISA kit (Catalog Number SL2183Hu) album of Sunlong Biotech CO., Ltd. (Hangzhou, China). According to the manufacturer’s protocol, each sample was incubated with the tracer from the kit for 2 h, following the instructions precisely to maintain intra-batch variation below 10% and inter-batch variation below 15%.

Co-founder factors: Mothers with potential confounding factors—such as systemic trauma, sepsis, or intense physical activity—were excluded from the study to minimize bias and ensure a more homogenous sample. As a result, these variables were not included in the data collection or analysis, given their limited relevance to the study population.

Arterial blood gas analysis: The blood gas analysis was performed from umbilical cord blood immediately drawn after the delivery. A sample of the blood was immediately inserted into a blood gas/electrolyte analyzing system. (COBAS B-221, ROSHE) for pH, pCO_2_, pO_2_, Hb, Hct, O_2_Hb, COHb, Ca^2+^, K^+^, Na^+^, CI^−^, SO_2_, BB, BE, HCO_3_, Osm measure and compared between the 2 groups, and the remaining blood sample was then used for the S100β study assay.

### 2.4. Statistical Analysis

The primary outcome of the study involved collecting blood samples for the analysis of brain injury biomarker S100β. These samples were obtained from the arterial line of maternal blood, both preoperative and postoperative, as well as from the umbilical artery of C-section deliveries performed under either spinal or general anesthesia. The secondary outcomes were invasive hemodynamic monitoring, surgery and anesthesia outcomes, umbilical cord blood gas values, Apgar scores, neonatal asphyxia rate, and maternal postoperative numeric pain rating scale (NPRS) compared in two groups. Categorical variables were expressed as frequencies and percentages. Continuous variables were assessed for distribution using the Shapiro–Wilk test, and dependent variables were normally distributed. Differences between the two groups were calculated using a parametric test. Differences between the means of the two groups were evaluated by the *t* test. Proportional differences between groups were calculated using the Chi-square test. In case of abnormal distribution, differences between the three groups were determined by Friedman’s test, and the Wilson Signed Rank test was used for differences between two groups.

## 3. Results

Sixty patients were recruited for the study. This study included a total of 60 pregnant women undergoing caesarean section, equally divided into two anesthesia groups: Spinal Anesthesia (SA) and General Anesthesia (GA) (n = 30 per group). To ensure comparability before assessing anesthetic effects on fetal outcomes, maternal demographic and obstetric characteristics were analyzed between the two groups. They were comparable among the study groups, with no statistically significant differences (*p* > 0.05), as shown in Table 1.

### 3.1. Maternal and Neonatal Brain Injury Marker and C Section Anesthesia Types

The mean serum S100β levels were 194.1 ± 45.8 ng/L in all women pre-operatively with no significant difference observed between the two groups (*p* = 0.231). Furthermore, there was no significant difference between the SA and GA group S100β protein levels post-surgery (*p* = 0.375) and in newborn blood serum (*p* = 0.143) (Table 2).

Within two hours following cesarean delivery, among neurological complications—including chills, headache, nausea, vomiting, weakness, and tinnitus—patients experiencing headache exhibited significantly elevated serum S100β protein levels compared to those without headache (*p* < 0.05) (Appendix A).

The changes in S100β protein levels before and after surgery were different for each group. The maternal blood S100β post-surgery and neonatal umbilical cord blood protein levels were different in the GA group (*p* = 0.047), but there was a difference in the SA group. In the SA group, preoperative S100β concentration in maternal blood was 195.1 ± 36.2 ng/L, then increased to 200.9 ± 42.9 ng/L at the end of operation. Also, in the GA group, preoperative S100β concentration in maternal blood was 193.0 ± 54.3 ng/L, then increased to 197.0 ± 42.7 at the end of operation (Figure 2).

### 3.2. Maternal Perioperative Outcome by Anaesthesia Groups

The perioperative outcomes, intraoperative time, drug dosage, and postoperative complications between the two groups are shown in Table 3 and Table 4. There were zero noteworthy differences in time from anesthesia to incision, the operation duration, infused crystalloid volume, blood loss, and urine output (Table 3). During anesthesia, most participants received normal saline and oxytocin, and there was no statistically significant difference in the average doses used between the two groups. However, the main difference between the two anesthesia methods lies in the route of drug administration and the specific medications used. We have specified the drugs used during each anesthesia method in the methodology section. No statistically significant association was found between the dosage of anesthetic agents used and S100β protein levels.

However, the frequency of headache and nausea was more common in the SA group (*p* < 0.0001, *p* < 0.0001) while hypertension and tachycardia were more frequent in the GA group (*p* = 0.079, *p* < 0.0001) during surgery. The occurrence of a headache may be related to the medications used during spinal anesthesia and requires further investigation. Since these headaches occurred immediately during surgery, they are unlikely to be related to post-puncture headaches. Also, although postoperative headache occurred in five cases in the spinal group and in one case in the general anesthesia group, the difference was not statistically significant. Postoperative weakness was significantly different in the SA and GA groups (*p* = 0.044). We detected that the time of onset of pain was significantly shorter in the GA group and 13 (43.3%) patients felt pain after surgery within ≤60 min. The NPRScore and length of stay in hospital between the two groups were also similar. (*p* = 0.105, *p* = 0.232) (Table 4).

### 3.3. Neonatal Outcomes by Anaesthesia Groups

No significant differences were found in Apgar scores and neonatal asphyxia rates between the two groups (*p* = 0.476) and blood gas outcomes of the umbilical artery (UA) in Table 5. It shows that pH and Ca^2+^ levels of UA were lower in the GA group (*p* = 0.009, *p* = 0.0001).

## 4. Discussion

The stress response to surgery triggers various physiological and biochemical changes, including sympathetic nervous system activation, the release of stress hormones such as cortisol and catecholamines, and modulation of immune function [25]. Exposure to anesthetic agents has been shown to induce programmed cell death (apoptosis) in glial and neuronal cells of the central nervous system, potentially leading to neurotoxicity and brain injury. Numerous animal and in vitro studies have reported that anesthetic agents can exert harmful effects on the developing brain [26], particularly barbiturates, ketamine, propofol, and inhaled anesthetics [27]. Given the potential variability of neuroinflammatory markers with different anesthesia types in pregnant women undergoing caesarean section [14,15], S100β—one of the most extensively studied biomarkers in this context—was selected as the primary marker to evaluate and compare the effects of general anesthesia and spinal anesthesia in the present study [28]. Therefore, S100β was chosen as the primary biomarker of interest in the present study. This study examined the brain injury biomarker S100β during cesarean section under general anesthesia (GA; propofol, fentanyl, isoflurane) and spinal anesthesia (SA; bupivacaine, fentanyl). Perioperative measurement of S100β levels, which reflects glial cell activity and possible neuroinjury, was performed before and after surgery. There was no statistically significant difference in S100β concentrations between the spinal and general anesthesia. Maternal arterial S100β concentrations showed similar patterns in both groups after caesarean section. This finding aligns with those of Zhendong Xu et al., who reported no significant change in maternal venous S100β levels following epidural or general anesthesia during C-section [29]. Some studies have explored the relationship between the dosage of anesthetic agents and S100β protein levels, although the findings are inconsistent. Results often vary depending on the type and dose of the anesthetic drug, the anesthesia technique, and the patient’s condition. The present study found no statistically significant correlation between the administered doses of anesthetic agents and postoperative S100β protein levels.

Additionally, our results indicated that cesarean delivery itself does not appear to significantly influence maternal S100β concentrations. However, some studies have reported higher S100β levels in vaginal deliveries compared to elective C-sections, suggesting delivery-mode dependency [29,30]. One previous study observed a significant decrease in the umbilical artery/umbilical vein (UA/UV) S100β ratio after GA compared to epidural anesthesia, while maternal venous S100β levels remained largely unchanged [31]. In our study, no statistically significant difference was observed in Apgar scores within the first 10 min between neonates delivered under general anesthesia and those under spinal anesthesia. This suggests that general anesthesia does not exert a detrimental effect on neonatal respiratory or hemodynamic function. Spinal and epidural anesthesia administered during vaginal delivery or caesarean section have not been associated with an increased risk of postpartum low back pain (LBPP). Although the potential relationship between epidural anesthesia and persistent postpartum pain remains a subject of debate, current evidence does not support a significant association. Furthermore, no statistically significant differences in C-section rates were observed across the various sub-groups [29].

We also established a reference range for serum S100β in third-trimester pregnant Mongolian women, identifying a baseline of 194.1 ± 45.8 ng/L. Notably, S100β levels measured beyond 37 weeks of gestation exceeded this normal range [32]. We propose two primary interpretations for our findings. Neither general nor spinal anesthesia caused detectable neuronal injury during cesarean delivery. This may be attributed to the short duration of surgery and, consequently, limited exposure to anesthetic agents. In fact, brief GA exposure might confer some neuroprotection under stress conditions [33], possibly explaining the slightly lower post-operative S100β levels in the GA group. The absence of maternal (e.g., CNS disorders), obstetric (e.g., diabetes, hypertension, placental insufficiency), or fetal (e.g., acute/chronic hypoxia) complications—conditions known to elevate S100β—may have influenced our results. Moreover, previous studies support the concept that fetal-origin S100β can pass into maternal circulation via a physiological gradient [34], consistent with our finding that UA S100β levels exceeded maternal concentrations. Although we did not find a significant association between S100β concentrations and perioperative variables, we observed moderate correlations with some umbilical artery blood gas parameters. In conclusion, our findings suggest that cesarean delivery under either anesthetic technique does not significantly alter maternal S100β levels, and thus may not be associated with acute brain injury. However, the anesthetic agents used could influence biomarker profiles.

Our study’s strengths include the use of a new biomarker that allows for a comprehensive evaluation of the effects of two anesthesia methods on both maternal and fetal brain damage during C-section. Firstly, this is the first known investigation to assess S100β levels in the maternal and umbilical circulation among pregnant Mongolian women undergoing cesarean section, thus contributing unique regional data to the global literature. Secondly, the study provides a direct comparison between general and spinal anesthesia, offering valuable insights into their respective impacts on neonatal biochemical outcomes and maternal neurobiological markers. Thirdly, by including both maternal and fetal (umbilical arterial) blood samples, we were able to explore the potential fetal origin of S100β more comprehensively. Additionally, the use of standardized anesthetic protocols and perioperative care ensured internal consistency, while the integration of neonatal Apgar scores and arterial blood gas analysis allowed for a broader evaluation of neonatal well-being. These elements collectively strengthen the scientific value and clinical relevance of our findings. But this study has several limitations. First, the relatively small sample size limits the statistical power and generalizability of our findings. Second, the study population consisted exclusively of pregnant women from a single tertiary hospital in Mongolia’s capital city, further restricting external applicability. Third, the inclusion of non-anesthetized, vaginally delivering women as a control group would have allowed for a more rigorous assessment of the impact of anesthesia on neonatal and maternal biomarkers. The absence of blinding during participant recruitment and the lack of established baseline S100β concentrations in both non-pregnant women and those who underwent vaginal delivery without anesthesia limited our ability to contextualize and generalize the findings. Additionally, the lack of long-term follow-up prevents conclusions regarding the potential delayed neurological effects of anesthetic exposure.

## 5. Conclusions

In conclusion, S100β concentrations slightly increased after C-section in both the SA and GA groups. The form of anesthesia seems to be irrelevant for the brain injury marker of S100β level. However, further research is needed to confirm these findings and fully evaluate any potential long-term effects.

## Figures and Tables

**Figure 1 medicina-61-01382-f001:**
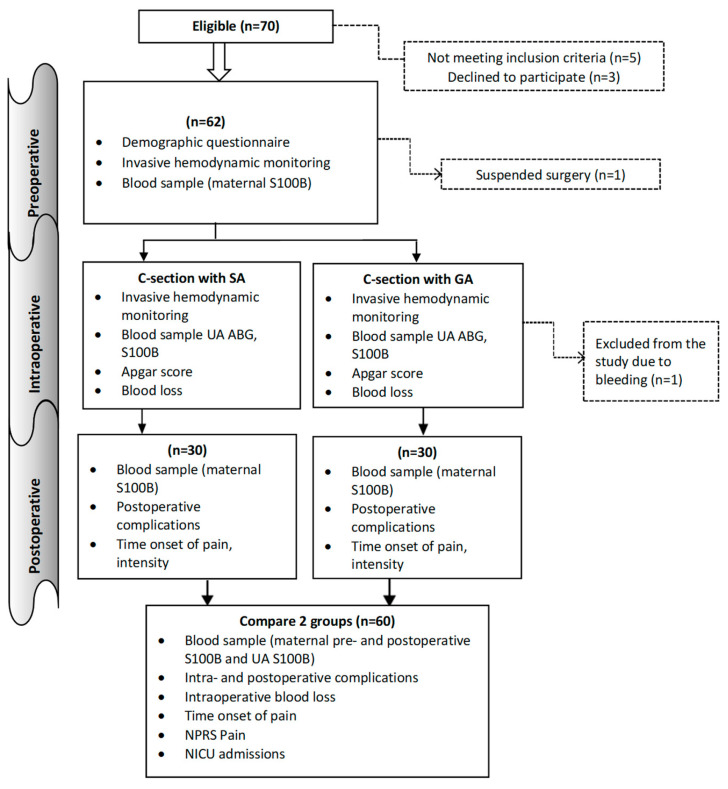
Flowchart detailing the study. GA: General anesthesia, SA: Spinal anesthesia, ABG: Arterial blood gas, UA: Umbilical artery, NPRS: Numeric pain rating scale, NICU: Neonatal intensive care.

**Figure 2 medicina-61-01382-f002:**
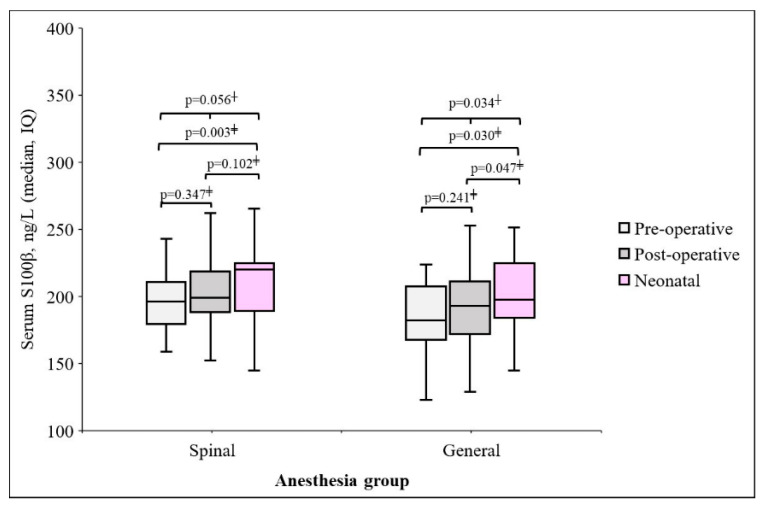
Changes in maternal and neonatal S100β protein by each anesthesia group, ^†^ Repeated measure ANOVA, ^‡^ Wilcoxon signed rank test.

**Table 1 medicina-61-01382-t001:** Participants’ general characteristics by anesthesia group (n = 60).

Variables		Anesthesia Groups				*p* Value	Total	
		SA Group		GA Group				
		n	%	n	%		n	%
Maternal age, y, mean ± std		30.1 ± 5.0		31.2 ± 5.1		0.302 ^†^	30.6 ± 5.1	
Weight, kg, mean ± std		69.6 ± 9.7		70.0 ± 18.3		0.203 ^†^	69.8 ± 14.5	
Height, cm, mean ± std		160.2 ± 3.4		156.4 ± 17.9		0.519 ^†^	158.3 ± 12.9	
Location (Urban)		16	53.3	14	46.7	0.606	30	50.0%
Education status						0.375		
	<High school degree	3	10.0	2	6.7		5	8.3%
	High school graduate	7	23.3	12	40.0		19	31.7%
	≥Bachelor’s degree	20	66.7	16	53.3		36	60.0%
Working conditions						0.601		
	Normal	28	93.3	29	96.7		57	95.0%
	Abnormal	2	6.6	0	0.0		3	5.0%
Abdominal surgery, yes		13	43.3	13	44.8	0.908	26	44.1%
Allergy, yes		3	10.0	2	6.7	0.640	5	8.3%
Gestation age at 1st prenatal visit						0.583		
	≤8 weeks	20	66.7	17	56.7		37	61.7%
	8–12 weeks	8	26.7	9	30.0		17	28.3%
	12–16 weeks	2	6.7	2	6.7		4	6.7%
	≥16 weeks	0	0.0	2	6.7		2	3.3%
Gestational age						0.633		
	37–38 weeks	8	26.7	5	16.7		13	21.7%
	38–39 weeks	10	33.3	12	40.0		22	36.7%
	39–40 weeks	12	40.0	13	43.3		25	41.7%
Previous Births						0.372		
	1–2 births	15	50.0	13	43.3		28	46.7%
	3–4 births	14	46.7	13	43.3		27	45.0%
	5≤ births	1	3.3	4	13.3		5	8.3%

^†^ Group differences were compared using Student *t* test. Values are mean (SD), number of observations (n), and 95% confidence intervals. SA: spinal anesthesia, GA: general anesthesia.

**Table 2 medicina-61-01382-t002:** Brain injury markers in 2 groups (n = 60).

Variables		Groups		*p* Value ^‡^	Total
		Spinal	General		
		Mean, Std	Mean, Std		
S100β, ng/L					
	Pre-operative	195.1 ± 36.2	193.0 ± 54.3	0.231	194.1 ± 45.8
	Post-operative	200.9 ± 42.9	197.0 ± 42.7	0.375	198.9 ± 42.5
	Umbilical cord blood	221.2 ± 52.8	203.1 ± 60.6	0.143	214.9 ± 57.1

^‡^—Wilcoxon signed rank test.

**Table 3 medicina-61-01382-t003:** Surgery time and drug doses by study groups (n = 60).

Variables	Groups				*p* Value	Total	
	SA		GA				
	Mean	Std	Mean	Std		Mean	Std
Surgery time (min)	49.9	8.0	51.6	10.4	0.471	50.7	9.2
Dosage of drugs							
Fentanyl dose (mcg)	14.3	2.2	231.7	58.8	0.0001 *	123.0	117.1
Succinylcholine (mg)			104.7	12.5		104.7	12.5
Tracrium dose (mg)			43.1	40.2		43.1	40.2
Bupivacaine dose (mg)	10.3	0.5				10.3	0.5
Ephedrine dose (mg)	17.9	9.0				17.9	9.0
Oxytocin dose (iu)	21.5	3.5	22.8	3.1		22.2	3.4
Isotonic solution (mL)	1563.3	237.1	1561.7	235.9	0.972	1562.5	234.5

Std—Standard deviations, number of observations (n), and 95% confidence intervals, * Significant difference between 2 groups.

**Table 4 medicina-61-01382-t004:** Caesarean Surgery Intraoperative and Postoperative Complications by Anesthesia Method.

Variables		Groups				*p* Value	Total	
		SA		GA				
		n	%	n	%		n	%
Complications during surgery								
	Hypotension	5	16.7%	1	3.3%	0.085	6	10.0%
	Hypertension	0	0.0%	12	40.0%	0.0001	12	20.0%
	Bradycardia	0	0.0%	2	6.7%	0.0001	2	3.3%
	Tachycardia	5	16.7%	11	36.7%	0.079	16	26.7%
	Nausea	15	50.0%	0	0.0%	0.0001	15	25.0%
	Headache	2	6.7%	0	0.0%	0.0001	2	3.3%
Complications after surgery								
	Nausea	11	36.7%	9	30.0%	0.075	20	33.3%
	Vomiting	2	6.7%	1	3.3%	0.553	3	5.0%
	Headache	5	16.7%	1	3.3%	0.085	6	10.0%
	Shivering	4	13.3%	3	10.0%	0.677	7	11.7%
	Weakness	1	3.3%	6	20.0%	0.044	7	11.7%
	Tinnitus	3	10.0%	0	0.0%	0.237	3	5.0%
	Shortness of breath	4	13.3%	2	6.7%	0.671	6	10.0%
Time of onset of pain						0.0001		
	≤60 min	1	3.3%	13	43.3%		14	23.3%
	61–119 min	16	53.3%	14	46.7%		30	50.0%
	120–179 min	13	43.3%	2	6.7%		15	25.0%
	≥180 min	0	0.0%	1	3.3%		1	1.7%
Blood loss during surgery, mL (mean, std)		431.7 ± 72		456.0 ± 83		0.105	443.8 ± 78	
Urine output, mL (mean, std)		155.0 ± 95		195.0 ± 93		0.232	175.0 ± 78	

**Table 5 medicina-61-01382-t005:** Neonatal Arterial Blood Gas Profiles and Clinical Outcomes Following Spinal versus General Anesthesia (n = 60).

ABG Values		Total		Groups				*p* Value
				SA		GA		
		Mean	SD	Mean	SD	Mean	SD	
pH		7.30	0.04	7.31	0.04	7.28	0.04	0.009
pCO_2_ (mmHg)		42.8	8.0	41.3	9.6	44.3	5.9	0.157
PO_2_ (mmHg)		48.2	29.2	45.5	27.3	51.0	31.2	0.476
Hb (g/L)		146.5	28.1	142.2	37.0	150.9	14.2	0.237
Hct (%)		44.3	6.6	43.8	8.6	44.8	3.8	0.588
O_2_Hb (%)		73.1	15.9	70.1	14.6	76.1	16.9	0.146
HHb (%)		23.2	15.6	25.0	14.6	21.5	16.7	0.385
MetHb (%)		0.8	0.1	0.8	0.1	0.9	0.2	0.389
Ca^2+^ (mmol/L)		1.41	0.15	1.34	0.15	1.47	0.12	0.0001
K^+^ (mmol/L)		5.80	1.74	5.91	1.93	5.70	1.55	0.643
Na^+^ (mmol/L)		134.9	5.0	133.5	6.7	136.3	1.8	0.029
Cl^−^ (mmol/L)		103.5	2.0	103.2	2.1	103.8	1.9	0.263
SO_2_ (%)		75.3	16.7	72.6	16.1	78.0	17.2	0.216
BB (mmol/L)		42.1	2.0	42.1	2.2	42.1	1.8	0.923
BE (mmol/L)		−5.9	1.9	−5.8	2.2	−5.9	1.6	0.864
ctCO_2_		18.3	2.0	18.0	2.3	18.5	1.5	0.341
HCO_3_^−^		20.4	2.1	20.1	2.4	20.7	1.7	0.263
Osm (mOsm/kg)		271	4	269	5	272	3	0.004
Hospital length (d)		4	1	4.1	0.9	3.6	0.5	0.351
Apgar score, n,%								0.476
	4–6 score	12	20.3	7	24.1%	5	16.7%	
	7 < score	47	79.7	22	75.9%	25	83.3%	

*p* value is calculate Student *t* test.

## Data Availability

The datasets used and/or analyzed during the current study are available from the corresponding author on reasonable request.

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
