# Peer review of "A Comparative Study of Brain Injury Biomarker S100β During General and Spinal Anesthesia for Caesarean Delivery: A Prospective Study"

_medicina, 2025, doi:10.3390/medicina61081382_

Round 1
Reviewer 1 Report
Comments and Suggestions for Authors
The study is interesting and well written, presenting valuable insight into the neuroinflammatory response in parturient women, particularly through the assessment of S100β levels following general (GA) and spinal anesthesia (SA) during and after cesarean. The observed increase in S100β levels post-GA compared to SA is eye-catching and merits further discussion to enhance the impact and readership of the work. The authors are encouraged to explore
- Is there any correlation between postoperative S100β levels and neurological symptoms (e.g., headache, cognitive dysfunction)?
- What is the temporal pattern of S100β release post-anesthesia (e.g., immediately post-op, 6 hours, 24 hours)?
- How can author rule out that the difference observed in GA and SA after c section is not due to difference in drugs administered during surgery.
- Authors are encourage to discussion other neuroinflammatory markers that are expected to change in GA and SA in pregnant women.
- Authors should discuss the possibility of back ache after epidural? Patients reports severe back aches after epidural.
Author Response
Comments 1: Is there any correlation between postoperative S100β levels and neurological symptoms (e.g., headache, cognitive dysfunction)? Thank you for pointing this out. We agree with this comment. (line nimbers 241-245).
"Within two hours following Cesarean delivery, among neurological complications—including chills, headache, nausea, vomiting, weakness, and tinnitus—patients experiencing headache exhibited significantly elevated serum S100β protein levels compared to other symptoms maternals (p < 0.05)."
Comments 2: What is the temporal pattern of S100β release post-anesthesia (e.g., immediately post-op, 6 hours, 24 hours)? Agree. We have accepted your suggestion and add it in the abstract, introduction section. We have modified and add to emphasize this point. (Line numbers 93-99)
S100β typically within 30 minutes to 2 hours after brain injury or surgery. S100β has a short half-life (~30–120 minutes), so sustained elevation beyond 6–12 hours may indicate ongoing or secondary injury.
Comments 3: How can author rule out that the difference observed in GA and SA after c section is not due to difference in drugs administered during surgery. (Line 258-264)
Response 3: Agree. We have modified and add to emphasize this point. We have accepted your suggestion and incorporated the relevant information into the methodology, results, and conclusion sections. To control for the effects of medications, we excluded patients who were using drugs known to cause brain injury with prolonged use. During anesthesia, most participants received normal saline and oxytocin, with no statistically significant difference in the mean doses between the two groups. The primary difference between the two anesthesia methods lies in the route of drug administration and the specific medications used. We have detailed the drugs administered for each anesthesia method in the methodology section. Furthermore, the differences in medication between the two methods have been emphasized in the discussion and conclusion sections. |
Comments 4: Authors are encourage to discussion other neuroinflammatory markers that are expected to change in GA and SA in pregnant women. (286-290) Response 4: Agree. We have modified and add to emphasize this point. We have accepted your suggestion and add it in the discussion section regarding other biomarkers. Given the potential variability of neuroinflammatory markers with different anaesthe-sia types in pregnant women undergoing caesarean section [15,29], S100B—one of the most extensively studied biomarkers in this context—was selected as the primary marker to evaluate and compare the effects of general anaesthesia (GA) and spinal an-aesthesia (SA) in the present study. |
Comments 5: Authors should discuss the possibility of back ache after epidural? Patients reports severe back aches after epidural. Thank you for pointing this out. We agree with this comment and added it in discussion section.(314-319) ]. In our study, no statistically significant difference was observed in Apgar scores within the first 10 minutes between neonates delivered under general anesthesia and those under spinal anesthesia. This suggests that general anesthesia does not exert a detri-mental effect on neonatal respiratory or hemodynamic function. Spinal and epidural anaesthesia administered during vaginal delivery or caesarean section (CS) have not been associated with an increased risk of postpartum low back pain (LBPP). Although the potential relationship between epidural anaesthesia and persistent postpartum pain remains a subject of debate, current evidence does not support a significant association. Furthermore, no statistically significant differences in CS rates were observed across the various sub-groups |
Response 5: Spinal and epidural anaesthesia administered during vaginal delivery or caesarean section (CS) have not been associated with an increased risk of postpartum low back pain (LBPP). Although the potential relationship between epidural anaesthesia and persistent postpartum pain remains a subject of debate, current evidence does not support a significant association. Furthermore, no statistically significant differences in CS rates were observed across the various sub-groups. |

Reviewer 2 Report
Comments and Suggestions for Authors
The topic was exploring S100β in relation to anesthetic technique during cesarean section, and is still timely and relevant. The prospective design and assessment of both maternal and umbilical samples are strengths. But, However, there are some key aspects which require some improvements:
- Introduction: Please clarify the knowledge gap and highlight the study’s novelty and rationale.
- Methods: Important details are lacking, including sample timing, group allocation, assay validation, and intraoperative variable control
- Discussion: Expand interpretation with relevant literature and biological context. Address more limitations beyond sample size.
- Conclusion: Avoid overstatements and align conclusions closely with your data. Suggest concrete directions for future studies.
With these revisions, i think the manuscript has potential to make a meaningful contribution in obstetric anesthesia filed.

Author Response
Comments 1: The background lacks a clear rationale connecting anesthesia, potential brain injury, and the selection of S100β as a biomarker, especially in the context of cesarean section (CS). A concise statement justifying this connection would enhance clarity. (line 6-10) Response 1: We have agree, We have tried to include the rationale more clearly in the summary. “S100β is a well-established biomarker of brain injury and blood–brain barrier disruption, and its prolonged elevation beyond 6–12 hours—despite a short half-life—may indicate ongoing neuronal injury. Its use in cesarean section (CS) remains limited, despite the potential neurological implications of both surgical stress and anesthetic technique.” |
|
Comments 2: The objective does not explicitly state that the study aims to compare maternal and neonatal S100β levels between general and spinal anesthesia groups. (line 10-12) Response 2: Agree. We have change and add study objectives. “This study evaluates potential brain injury during Cesarean section by comparing maternal and neonatal S100β levels under general and spinal anesthesia.” |
|
Comments 3: The methods section omits inclusion and exclusion criteria, as well as specific data points to be collected. (line 12-19) “This observational prospective study compared changes in the S100β brain damage biomarker in maternal(pre and post-surgery) and umbilical artery blood during elective c-sections under general or spinal anaesthesia. The 60 parturient women underwent a C-section from July 1, 2021 to Dec 30, 2023 had been evenly distributed into 2 groups General anaesthesia (n=30), and Spinal anaesthesia group (n=30). It included healthy term pregnant women((aged 18–40, ASA I–II)) and excluded those with major comorbidities or emergency conditions” |
|
Comments 4: The conclusion claims safety and absence of harm (“no evidence suggesting harmful effects”) too strongly. Such assertions should be grounded in the presented results and stated more cautiously. (line 366-369) Response 4: Agree. Thank you for your valuable advices. We have modified and included the conclusion section. “. In this study, spinal and general anaesthesia did not appear to adversely affect maternal or fetal health during caesarean section when administered appropriately. However, further studies are needed to confirm these findings and fully evaluate any potential long-term impacts.” |
|
Response 5: Thank you for your very clear and valuable advice. We have included it in the introduction section. “The choice of anesthetic technique is often guided by clinical circumstances and patient factors, yet emerging evidence suggests it may also have significant implications for neuroinflammatory responses and glial activation. Understanding the neuroglial impact of different anesthetic modalities is especially important given their potential to influ-ence postoperative cognitive outcomes, neurodevelopment, and neurodegeneration. By elucidating how general and regional anesthesia affect neuroimmune signaling, this study aims to inform safer anesthetic practices and identify modifiable risk factors relevant to perioperative brain health.” |
|
Comments 6: Adding local context (e.g., CS practice in Mongolia) could improve its significance.(line 110-112) Response: Agree, We have accepted the suggestion and added it. “In Mongolia, spinal anesthesia (SA) and general anesthesia (GA) are the predominant techniques used during Cesarean section. However, to date, no studies have systematically examined their respective effects on maternal and fetal outcomes in this context.” |
|
Response 7. Agree, We have accepted the suggestions and add in introduction. “Several neuroinflammatory markers—such as S100B, neuron-specific enolase (NSE), glial fibrillary acidic protein (GFAP), tumor necrosis factor-alpha (TNF-α), and interleukin-6 (IL-6)—have been investigated or are hypothesized to vary in response to general anesthesia (GA) and spinal anesthesia (SA), particularly in pregnant women undergoing Cesarean section. Among these markers, S100B has been most frequently studied in relation to Cesarean delivery and comparisons between anesthesia modalities. Therefore, S100B was chosen as the primary biomarker of interest in the present study.” |
|
Comment 8. A clear statement explaining why comparing the effects of anesthesia on S100β levels is both clinically and scientifically relevant is needed.( line 95-99) Response 8. We have accepted the suggestions and add in introduction. “This protein has been shown to be valuable in animal models for detecting general anesthesia-mediated acute neurologic damage in the developing brain23, and also used to monitor the impact of prenatal drug exposure (24-25). However, its effectiveness and usefulness to reflect anesthesia associated brain damage in human subjects is yet to be confirmed.” |
|
Comment 9. The rationale for measuring neonatal S100β levels should be explicitly addressed—what is the clinical or physiological significance?(line 102-109) Response 9. We have agree this comment and add in introduction. “Due to its high sensitivity in detecting fetal brain damage caused by asphyxia or prenatal stress, S100β is considered a strong candidate for investigation in related clinical studies. Some studies suggest that S100β may also serve as a biomarker for neurodegeneration in the developing brain induced by general anesthesia. This is supported by the observation that S100β levels remain consistent across umbilical artery (UA), umbilical vein (UV), and the UA/UV ratio, indicating its reliability as a peripheral biomarker in patients undergoing different types of anesthesia.” |
|
Comment 10. Clarify the relevance of neonatal outcomes in relation to maternal anesthesia. (line 310-314) Response 10. We have accepted the suggestions and added in discussion. In our study, no statistically significant difference was observed in Apgar scores within the first 10 minutes between neonates delivered under general anesthesia and those under spinal anesthesia. This suggests that general anesthesia does not exert a detrimental effect on neonatal respiratory or hemodynamic function. |
|
Comment 11. The study design remains unclear: is it a randomized controlled trial or an observational comparison?(Line numbers 121) Response 11. We have removed the definitions of controlled and single-center and clarified that it is an observational study in which one participant was measured three times. So we clarified that we followed up/longitudinal. “The study was observational a prospective, longitudinal design.” |
|
Response 13. We have taken your suggestion into account and included the outcome measurement indicators in the methodology section. · Maternal: demographic data, intraoperative hemodynamics, operative duration, blood loss, postoperative symptoms (PONV, headache, fever, pain). (line 158-173) “Preoperative and intraoperative: Preoperative assessments were carried out by main in-vestigators, the researchers gathered demographics data and basal health question-naires. Radial arterial cannulation was performed for all the patients using 20G Sur-flo® (Terumo China holding Co., Ltd.) under local anaesthesia (2% lidocaine), and ar-terial blood pressure was monitored. Invasive blood pressure readings including sys-tolic, diastolic, and mean blood pressures were obtained. Heart rate and SpO2 were measured by the fingertip photoelectric sensor (manufactured: Guangzhou Sichuang Hongyi Electronic Technology Co., Ltd.). Time from anaesthesia induction to delivery, total operative time, intraoperative blood loss, urine output, and complications during surgery were recorded in both groups. All vital signs were recorded. For the two hours of the postoperative stage, patients were monitored in the recovery room. Complications after C-section within two hours were recorded in both groups including postoperative nausea, vomiting, headache, fever, and pain. The patients were assessed using the Numeric pain rating scale (NPS), which showed the severity of the postoperative level of pain from a scale of 0 to 10, with 0 = no pain and 10 being the worst pain ever. The onset time was also recorded.” · Neonatal: Apgar scores, NICU admission, arterial blood gas parameters, electrolytes, hemoglobin derivatives. (line 174-185) Apgar scores, umbilical arterial blood gas parameters, electrolyte levels, hemoglobin derivatives, incidence of neonatal asphyxia, and neonatal intensive care unit (NICU) admission data were collected for the neonates. “Following the delivery of the baby, pediatricians assessed the neonatal condition, assigned Apgar scores at one and five minutes, and determined the need for admission to the Neonatal intensive care unit (NICU). An Apgar score of 7 to 10 was considered normal; 4 to 6, mild neonatal asphyxia; and 3 and below, severe neonatal asphyxia. The Apgar score was assessed based on 5 criteria: activity, pulse rate, grimace (reflex irritability), skin color, and respiratory effort. Following the delivery of the baby, the cord was cut between the two clamps placed approximately 10 to 12 cm apart and away from the placenta and newborn. One to three ml of blood was collected from the umbilical artery from between the clamps immediately following placental delivery. This protocol was designed to prevent contamination of S100β levels in the cord arterial blood by placental S100β, ensuring that the measured concentrations accurately reflect those originating from the newborn.”
|
|
Comment 14. While S100β is a sensitive marker of glial injury, it is not specific. Known confounders (e.g., systemic trauma, sepsis, intense physical activity) must be acknowledged and, ideally, measured. (line 203-206) Response 14. Mothers with potential confounding factors—such as systemic trauma, sepsis, or intense physical activity—were excluded from the study to minimize bias and ensure a more homogenous sample. As a result, these variables were not included in the data collection or analysis, given their limited relevance to the study population. |
|
Comment 16. The timing of sample collection (e.g., pre-anesthesia, post-SC, post-delivery) should be clearly described. (line 188-191) |
|
Comment 17. Interpretation of values must be aligned with validated reference ranges specific to sample type and population (maternal vs fetal).
|
|
Comment 18. The comparison between maternal and neonatal values from different blood compartments must be justified, as their physiological baselines differ. |
|
Comment 20. The manuscript currently underreports potential limitations. Agree. We have accepted the suggestions and added in discussion.(line 355-360) |
|
Third, the inclusion of non-anesthetized, vaginally delivering women as a control group would have allowed for a more rigorous assessment of the impact of anesthesia on neonatal and maternal biomarkers. |
|
Bias due to selection, lack of blinding, or missing baseline data may influence interpretation. We have accepted the suggestions and added in discussion. |
|
Comment 21. Numerous confounders affecting S100β levels were neither controlled nor statistically adjusted. (line 355-357) |
|
Bias due to selection, lack of blinding, or missing baseline data may influence interpretation. We have accepted the suggestions and added in discussion. |
|
Response 21. The absence of blinding during participant recruitment and the lack of established baseline S100β concentrations in both non-pregnant women and those who underwent vaginal delivery without anesthesia limited our ability to contextualize and generalize the findings. Comment 22. Addressing these in the discussion would significantly improve the study’s transparency and redibility. Response 22. Thank you for your valuable suggestion. We have carefully incorporated a detailed discussion of these points in the revised manuscript to enhance both the transparency and credibility of our study.
|
